# Federated Hyperparameter Tuning: Challenges, Baselines, and Connections to Weight-Sharing

**Mikhail Khodak, Renbo Tu, Tian Li**
Carnegie Mellon University
`{khodak,renbo,tianli}@cmu.edu`

**Liam Li**
Hewlett Packard Enterprise
`me@liamcli.com`

**Maria-Florina Balcan, Virginia Smith**
Carnegie Mellon University
`ninamf@cs.cmu.edu,smithv@cmu.edu`

**Ameet Talwalkar**
Carnegie Mellon University & Hewlett Packard Enterprise
`talwalkar@cmu.edu`

## Abstract

Tuning hyperparameters is a crucial but arduous part of the machine learning pipeline. Hyperparameter optimization is even more challenging in federated learning, where models are learned over a distributed network of heterogeneous devices; here, the need to keep data on device and perform local training makes it difficult to efficiently train and evaluate configurations. In this work, we investigate the problem of federated hyperparameter tuning. We first identify key challenges and show how standard approaches may be adapted to form baselines for the federated setting. Then, by making a novel connection to the neural architecture search technique of *weight-sharing*, we introduce a new method, FedEx, to accelerate federated hyperparameter tuning that is applicable to widely-used federated optimization methods such as FedAvg and recent variants. Theoretically, we show that a FedEx variant correctly tunes the on-device learning rate in the setting of online convex optimization across devices. Empirically, we show that FedEx can outperform natural baselines for federated hyperparameter tuning by several percentage points on the Shakespeare, FEMNIST, and CIFAR-10 benchmarks—obtaining higher accuracy using the same training budget.

## 1 Introduction

Federated learning (FL) is a popular distributed computational setting where training is performed locally or privately [30, 36] and where hyperparameter tuning has been identified as a critical problem [18]. Although general hyperparameter optimization has been the subject of intense study [3, 16, 26], several unique aspects of the federated setting make tuning hyperparameters especially challenging. However, to the best of our knowledge there has been no dedicated study on the specific challenges and solutions in federated hyperparameter tuning. In this work, we first formalize the problem of hyperparameter optimization in FL, introducing the following three key challenges:

1. **Federated validation data:** In federated networks, as the validation data is split across devices, the entire dataset is not available at any one time; instead a central server is given access to some number of devices at each communication round, for one or at most a few runs of local training and validation. Thus, because the standard measure of complexity in FL is the number of communication rounds, computing validation metrics exactly dramatically increases the cost.

35th Conference on Neural Information Processing Systems (NeurIPS 2021).

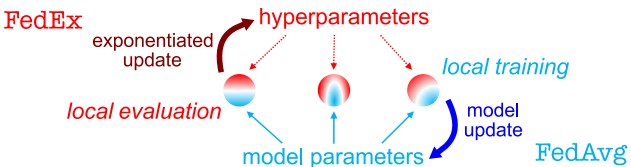

Figure 1: `FedEx` can be applied to any local training-based FL method, e.g. `FedAvg`, by interleaving standard updates to model weights (computed by aggregating results of local training) with exponentiated gradient updates to hyperparameters (computed by aggregating results of local validation).

2. **Extreme resource limitations:** FL applications often involve training using devices with very limited computational and communication capabilities. Furthermore, many require the use of privacy techniques such as differential privacy that limit the number times user data can be accessed. Thus we cannot depend on being able to run many different configurations to completion.

3. **Evaluating personalization:** Finally, even with non-federated data, applying common hyperparameter optimization methods to standard personalized FL approaches (such as finetuning) can be costly because evaluation may require performing many additional training steps locally.

With these challenges[1] in mind, we propose reasonable baselines for federated hyperparameter tuning by showing how to adapt standard non-federated algorithms. We further study the challenge of noisy validation signal due to federation, and show that simple state-estimation-based fixes do not help.

Our formalization and analysis of this problem leads us to develop `FedEx`, a method that exploits a novel connection between hyperparameter tuning in FL and the weight-sharing technique widely used in neural architecture search (NAS) [4, 34, 40]. In particular, we observe that weight-sharing is a natural way of addressing the three challenges above for federated hyperparameter tuning, as it incorporates noisy validation signal, simultaneously tunes and trains the model, and evaluates personalization as part of training rather than as a costly separate step. Although standard weight-sharing only handles architectural hyperparameters such as the choice of layer or activation, and not critical settings such as those of local stochastic gradient descent (SGD), we develop a formulation that allows us to tune most of these as well via the relationship between local-training and fine-tuning-based personalization. This make `FedEx` a general hyperparameter tuning algorithm applicable to many local training-based FL methods, e.g. `FedAvg` [36], `FedProx` [31], and `SCAFFOLD` [19].

In Section 4, we next conduct a theoretical study of `FedEx` in a simple setting: tuning the client step-size. Using the ARUBA framework for analyzing meta-learning [20], we show that a variant of `FedEx` correctly tunes the on-device step-size to minimize client-averaged regret by adapting to the intrinsic similarity between client data. We improve the convergence rate compared to some past meta-learning theory [20, 25] while not depending on knowing the (usually unknown) task-similarity.

Finally, in Section 5, we instantiate our baselines and `FedEx` to tune hyperparameters of `FedAvg`, `FedProx`, and `Reptile`, evaluating on three standard FL benchmarks: Shakespeare, FEMNIST, and CIFAR-10 [5, 36]. While our baselines already obtain performance similar to past hand-tuning, `FedEx` further surpasses them in most settings examined, including by 2-3% on Shakespeare.

**Related Work** To the best of our knowledge, we are the first to systematically analyze the formulation and challenges of hyperparameter optimization in the federated setting. Several papers have explored limited aspects of hyperparameter tuning in FL [7, 23, 38], focusing on a small number of hyperparameters (e.g. the step-size and sometimes one or two more) in less general settings (studying small-scale problems or assuming server-side validation data). In contrast our methods are able to tune a wide range of hyperparameters in realistic federated networks. Some papers also discussed the challenges of finding good configurations while studying other aspects of federated training [41]. We argue that it is critical to properly address the challenges of federated hyperparameter optimization in practical settings, as we discuss in detail in Section 2.

Methodologically, our approach draws on the fact that local training-based methods such as `FedAvg` can be viewed as optimizing a surrogate objective for personalization [20], and more broadly leverages the similarity of the personalized FL setup and initialization-based meta-learning [6, 11, 17, 25]. While `FedEx`'s formulation and guarantees use this relationship, the method itself is general-purpose

---

[1]A further challenge we do *not* address is that of the time-dependency of federated evaluation, c.f. [9].

and applicable to federated training of a single global model. Many recent papers address FL personalization more directly [13, 29, 35, 43, 47]. This connection and our use of NAS techniques also makes research connecting NAS and meta-learning relevant [10, 33], but unlike these methods we focus on tuning *non*-architectural parameters. In fact, we believe our work is the first to apply weight-sharing to regular hyperparameter search. Furthermore, meta-learning does not have the data-access and computational restrictions of FL, where such methods using the DARTS mixture relaxation [34] are less practical. Instead, FedEx employs the lower-overhead stochastic relaxation [8, 28], and its exponentiated update is similar to the recently proposed GAEA approach for NAS [27]. Running NAS itself in federated settings has also been studied [12, 15, 46]; while our focus is on non-architectural hyperparameters, in-principle our algorithms can also be used for federated NAS.

Theoretically, our work makes use of the average regret-upper-bound analysis (ARUBA) framework [20] to derive guarantees for learning the initialization, i.e. the global model, while simultaneously tuning the step-size of the local algorithm. The step-size of gradient-based algorithms has also been tuned on its own in the settings of data-driven algorithm design [14] and of statistical learning-to-learn [45].

## 2   Federated Hyperparameter Optimization

In this section we formalize the problem of hyperparameter optimization for FL and discuss the connection of its personalized variant to meta-learning. We also review FedAvg [36], a common federated optimization method, and present a reasonable baseline approach for tuning its hyperparameters.

**Global and Personalized FL**   In FL we are concerned with optimizing over a network of heterogeneous clients $i = 1, \ldots, n$, each with training, validation, and testing sets $T_i$, $V_i$, and $E_i$, respectively. We use $L_S(\mathbf{w})$ to denote the average loss over a dataset $S$ of some $\mathbf{w}$-parameterized ML model, for $\mathbf{w} \in \mathbb{R}^d$ some real vector. For hyperparameter optimization, we assume a class of algorithms $\texttt{Alg}_a$ hyperparameterized by $a \in \mathcal{A}$ that use *federated access* to training sets $T_i$ to output some element of $\mathbb{R}^d$. Here by "federated access" we mean that each iteration corresponds to a *communication round* at which $\texttt{Alg}_a$ has access to a batch of $B$ clients[2] that can do local training and validation.

Specifically, we assume $\texttt{Alg}_a$ can be described by two subroutines with hyperparameters encoded by $b \in \mathcal{B}$ and $c \in \mathcal{C}$, so that $a = (b, c)$ and $\mathcal{A} = \mathcal{B} \times \mathcal{C}$. Here $c$ encodes settings of a local training algorithm $\texttt{Loc}_c$ that take a training set $S$ and initialization $\mathbf{w} \in \mathbb{R}^d$ as input and outputs a model $\texttt{Loc}_c(S, \mathbf{w}) \in \mathbb{R}^d$, while $b$ sets those of an aggregation $\texttt{Agg}_b$ that takes the initialization $\mathbf{w}$ and outputs of $\texttt{Loc}_c$ as input and returns a model parameter. For example, in standard FedAvg, $\texttt{Loc}_c$ is $T$ steps of gradient descent with step-size $\eta$ and $\texttt{Agg}_b$ takes a weighted average of the outputs of $\texttt{Loc}_c$ across clients; here $c = (\eta, T)$ and $b = ()$. As detailed in the appendix, many FL methods can be decomposed this way, including well-known ones such as FedAvg [36], FedProx [31], SCAFFOLD [19], and Reptile [39] as well as more recent methods [1, 2, 29]. Our analysis and our proposed FedEx algorithm will thus apply to all of them, up to an assumption detailed next.

Starting from this decomposition, the global hyperparameter optimization problem can be written as

$$\min_{a \in \mathcal{A}} \quad \sum_{i=1}^{n} |V_i| L_{V_i}(\texttt{Alg}_a(\{T_j\}_{j=1}^{n})) \tag{1}$$

In many cases we are also interested in obtaining a device-specific local model, where we take a model trained on all clients and finetune it on each individual client before evaluating. A key assumption we make is that the finetuning algorithm will be the same as the local training algorithm $\texttt{Loc}_c$ used by $\texttt{Alg}_a$. This assumption can be justified by recent work in meta-learning that shows that algorithms that aggregate the outputs of local SGD can be viewed as optimizing for personalization using local SGD [20]. Then, in the personalized setting, the tuning objective becomes

$$\min_{a = (b,c) \in \mathcal{A}} \quad \sum_{i=1}^{n} |V_i| L_{V_i}(\texttt{Loc}_c(T_i, \texttt{Alg}_a(\{T_j\}_{j=1}^{n}))) \tag{2}$$

Our approach will focus on the setting where the hyperparameters $c$ of local training make up a significant portion of all hyperparameters $a = (b, c)$; by considering the personalization objective we will be able to treat such hyperparameters as architectural and thus apply weight-sharing.

**Algorithm 1:** Successive halving algorithm (SHA) applied to personalized FL. For the non-personalized objective (1), replace $L_{V_{ti}}(\mathbf{w}_i)$ by $L_{V_{ti}}(\mathbf{w}_a)$. For random search (RS) with $N$ samples, set $\eta = N$ and $R = 1$.

---

**Input:** distribution $\mathcal{D}$ over hyperparameters $\mathcal{A}$,
      elimination rate $\eta \in \mathbb{N}$, elimination rounds
      $\tau_0 = 0, \tau_1, \ldots, \tau_R$
sample set of $\eta^R$ hyperparameters $H \sim \mathcal{D}^{[\eta^R]}$
initialize a model $\mathbf{w}_a \in \mathbb{R}^d$ for each $a \in H$
**for** *elimination round* $r \in [R]$ **do**
  **for** *setting* $a = (b, c) \in H$ **do**
    **for** *comm. round* $t = \tau_{r-1} + 1, \ldots, \tau_r$ **do**
      **for** *client* $i = 1, \ldots, B$ **do**
        send $\mathbf{w}_a, c$ to client
        $\mathbf{w}_i \leftarrow \mathrm{Loc}_c(T_{ti}, \mathbf{w}_a)$
        send $\mathbf{w}_i, L_{V_{ti}}(\mathbf{w}_i)$ to server
      $\mathbf{w}_a \leftarrow \mathrm{Agg}_b(\mathbf{w}_a, \{\mathbf{w}_i\}_{i=1}^B)$
      $s_a \leftarrow \sum_{i=1}^B |V_{ti}| L_{V_{ti}}(\mathbf{w}_i) / \sum_{i=1}^B |V_{ti}|$
  $H \leftarrow \{a \in H : s_a \leq \frac{1}{\eta}\text{-quantile}(\{s_a : a \in H\})\}$
**Output:** remaining $a \in H$ and associated model $\mathbf{w}_a$

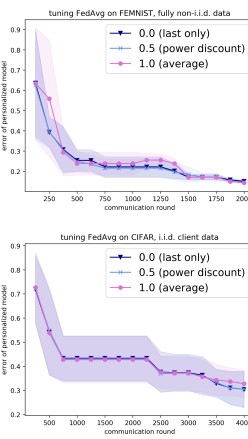

Figure 2: Tuning FL with SHA but making elimination decisions based on validation estimates using different discount factors. On both FEMNIST (top) and CIFAR (bottom) using more of the validation data does not improve upon just using the most recent round's validation error.

**Tuning FL Methods: Challenges and Baselines**    In the non-federated setting, the objective (1) is amenable to regular hyperparameter optimization methods; for example, a random search approach would repeatedly sample a setting $a$ from some distribution over $\mathcal{A}$, run $\mathrm{Alg}_a$ to completion, and evaluate the objective, saving the best setting and output [3]. With a reasonable distribution and enough samples this is guaranteed to converge and can be accelerated using early stopping methods [26], in which $\mathrm{Alg}_a$ is not always run to completion if the desired objective is poor at intermediate stages, or by adapting the sampling distribution using the results of previous objective evaluations [44]. As mentioned in the introduction, applying such methods to FL is inherently challenging due to

1. **Federated validation data:** Separating data across devices means we cannot immediately get a good estimate of the model's validation performance, as we only have access to a possibly small batch of devices at a time. This means that decisions such as which models to flag for early stopping will be noisy and may not fully incorporate all the available validation signal.

2. **Extreme resource limitations:** As FL algorithms can take a very long time to run in-practice due to the weakness and spotty availability of devices, we often cannot afford to conduct many training runs to evaluate different configurations. This issue is made more salient in cases where we use privacy techniques that only allow a limited number of accesses to the data of any individual user.

3. **Evaluating personalization:** While personalization is important in FL due to client heterogeneity, checking the performance of the current model on the personalization objective (2) is computationally intensive because computing may require running local training multiple times. In particular, while regular validation losses require computing one forward pass per data point, personalized losses require several forward-backward passes, making it many times more expensive if this loss is needed to make a tuning decision such as eliminating a configuration from consideration.

Despite these challenges, we can still devise sensible baselines for tuning hyperparameters in FL, most straightforward of which is to use a regular hyperparameter method but use validation data from a single round as a noisy surrogate for the full validation objective. Specifically, one can use random search (RS)—repeatedly evaluate random configurations—and a simple generalization called successive halving (SHA), in which we sample a set of configurations and partially run all of them for some number of communication rounds before eliminating all but the best $\frac{1}{\eta}$ fraction, repeating until only one configuration remains. Note both are equivalent to a "bracket" in Hyperband [26] and their adaptation to FL is detailed in Algorithm 1.

---

[2] For simplicity the number of clients per round is fixed, but all methods can be easily generalized to varying $B$.

As shown in Section 5, SHA performs reasonably well on the benchmarks we consider. However, by using validation data from one round it may make noisy elimination decisions, early-stopping potentially good configurations because of a difficult set of clients on a particular round. Here the problem is one of insufficient utilization of the validation data to estimate model performance. A reasonable approach to use more is to try some type of state-estimation: using the performance from previous rounds to improve the noisy measurement of the current one. For example, instead of using only the most recent round for elimination decisions we can use a weighted sum of the performances at all past rounds. To investigate this, we study a power decay weighting, where a round is discounted by some constant factor for each time step it is in the past. We consider factors 0.0 (taking the most recent performance only, as before), 0.5, and 1.0 (taking the average). However, in Figure 2 we show that incorporating more validation data this way than is used by Algorithm 1 by default does not significantly affect results.

Thus we may need a better algorithm to use more of the validation signal, most of which is discarded by using the most recent round's performance. We next propose `FedEx`, a new method that does so by using validation on each round to update a client hyperparameters distribution used to sample configurations to send to devices. Thus it alleviates issue (1) above by updating at each step, not waiting for an elimination round as in RS or SHA. By simultaneously training the model and tuning (client) hyperparameters, it also moves towards a fully single-shot procedure in which we only train once (we must still run multiple times due to server hyperparameters), which would solve issue (2). Finally, `FedEx` addresses issue (3) by using local training to both update the model and to estimate personalized validation loss, thus not spending extra computation on this more expensive objective.

## 3  Weight-Sharing for Federated Learning

We now present `FedEx`, a way to tune local FL hyperparameters. This section contains the general algorithm and its connection to weight-sharing; we instantiate it on several FL methods in Section 5.

**Weight-Sharing for Architecture Search**  We first review the weight-sharing approach in NAS, which for a set $\mathcal{C}$ of network configurations is often posed as the bilevel optimization

$$\min_{c \in \mathcal{C}} \ L_{\text{valid}}(\mathbf{w}, c) \quad \text{s.t.} \quad \mathbf{w} \in \arg\min_{\mathbf{u} \in \mathbb{R}^d} \ L_{\text{train}}(\mathbf{u}, c) \tag{3}$$

where $L_{\text{train}}, L_{\text{valid}}$ evaluate a single configuration with the given weights. If, as in NAS, all hyperparameters are architectural, then they are effectively themselves trainable model parameters [27], so we could instead consider solving the following "single-level" empirical risk minimization (ERM):

$$\min_{c \in \mathcal{C}, \mathbf{w} \in \mathbb{R}^d} \ L(\mathbf{w}, c) \quad = \quad \min_{c \in \mathcal{C}, \mathbf{w} \in \mathbb{R}^d} \ L_{\text{train}}(\mathbf{w}, c) + L_{\text{valid}}(\mathbf{w}, c) \tag{4}$$

Solving this instead of the bilevel problem (3) has been proposed in several recent papers [24, 27].

Early approaches to solving either formulation of NAS were costly due to the need for full or partial training of many architectures in a very large search space. The weight-sharing paradigm [40] reduces the problem to that of training a single architecture, a "supernet" containing all architectures in the search space $\mathcal{C}$. A straightforward way of constructing a supernet is via a "stochastic relaxation" where the loss is an expectation w.r.t. sampling $c$ from some distribution over $\mathcal{C}$ [8]. Then the shared weights can be updated using SGD by first sampling an architecture $c$ and using an unbiased estimate of $\nabla_{\mathbf{w}} L(\mathbf{w}, c)$ to update $\mathbf{w}$. The distribution over $\mathcal{C}$ may itself be adapted or stay fixed. We focus on the former case, adapting some $\theta$-parameterized distribution $\mathcal{D}_\theta$; this yields the stochastic relaxation objective

$$\min_{\theta \in \Theta, \mathbf{w} \in \mathbb{R}^d} \ \mathbb{E}_{c \sim \mathcal{D}_\theta} L(\mathbf{w}, c) \tag{5}$$

Since architectural hyperparameters are often discrete decisions, e.g. a choice of which of a fixed number of operations to use, a natural choice of $\mathcal{D}_\theta$ is as a product of categorical distributions over simplices. In this case, any discretization of an optimum $\theta$ of the relaxed objective (5) whose support is in the support of $\theta$ will be an optimum of the original objective (4). A natural update scheme here is exponentiated gradient [22], where each successive $\theta$ is proportional to $\theta \odot \exp(-\eta \tilde{\nabla})$, $\eta$ is a step-size, and $\tilde{\nabla}$ an unbiased estimate of $\nabla_\theta \mathbb{E}_{c \sim \mathcal{D}_\theta} L(\mathbf{w}, c)$ that can be computed using the re-parameterization trick [42]. By alternating this exponentiated update with the standard SGD update to $\mathbf{w}$ discussed earlier we obtain a simple block-stochastic minimization scheme that is guaranteed to converge, under certain conditions, to the ERM objective, and also performs well in practice [27].

**The** `FedEx` **Method**    To obtain `FedEx` from weight-sharing we restrict to the case of tuning only the hyperparameters $c$ of local training $\text{Loc}_c$.[3] Our goal then is just to find the best initialization $\mathbf{w} \in \mathbb{R}^d$ and local hyperparameters $c \in \mathcal{C}$, i.e. we replace the personalized objective (2) by

$$\min_{c \in \mathcal{C}, \mathbf{w} \in \mathbb{R}^d} \quad \sum_{i=1}^n |V_i| L_{V_i}(\text{Loc}_c(T_i, \mathbf{w})) \tag{6}$$

Note $\text{Alg}_a$ outputs an element of $\mathbb{R}^d$, so this new objective is upper-bounded by the original (2), i.e. any solution will be at least as good for the original objective. Note also that for fixed $c$ this is equivalent to the classic train-validation split objective for meta-learning with $\text{Loc}_c$ as the base-learner. More importantly for us, it is also in the form of the r.h.s. of the weight-sharing objective (4), i.e. it is a single-level function of $\mathbf{w}$ and $c$. We thus apply a NAS-like stochastic relaxation:

$$\min_{\theta \in \Theta, \mathbf{w} \in \mathbb{R}^d} \quad \sum_{i=1}^n |V_i| \mathbb{E}_{c \in \mathcal{D}_\theta} L_{V_i}(\text{Loc}_c(T_i, \mathbf{w})) \tag{7}$$

In NAS we would now set the distribution to be a product of categorical distributions over different architectures, thus making $\theta$ an element of a product of simplices and making the optimum of the original objective (6) equivalent to the optimum of the relaxed objective (7) as an extreme point of the simplex. Unlike in NAS, FL hyperparameters such as the learning rate are not extreme points of a simplex and so it is less clear what parameterized distribution $\mathcal{D}_\theta$ to use. Nevertheless, we find that crudely imposing a categorical distribution over $k > 1$ random samples from some distribution (e.g. uniform) over $\mathcal{C}$ and updating $\theta$ using exponentiated gradient over the resulting $k$-simplex works well. We alternate this with updating $\mathbf{w} \in \mathbb{R}^d$, which in a NAS algorithm involves an SGD update using an unbiased estimate of the gradient at the current $\mathbf{w}$ and $\theta$.

We call this alternating method for solving (7) `FedEx` and describe it for a general $\text{Alg}_a$ consisting of sub-routines $\text{Agg}_b$ and $\text{Loc}_c$ in Algorithm 2; recall from Section 2 that many FL methods can be decomposed this way, so our approach is widely applicable. `FedEx` has a minimal overhead, consisting only of the last four lines of the outer loop updating $\theta$. Thus, as with weight-sharing, `FedEx` can be viewed as reducing the complexity of tuning local hyperparameters to that of training a single model. Each update to $\theta$ requires a step-size $\eta_t$ and an approximation $\tilde{\nabla}$ of the gradient w.r.t. $\theta$; for the latter we obtain an estimate $\tilde{\nabla}_j$ of each gradient entry via the reparameterization trick, whose variance we reduce by subtracting a baseline $\lambda_t$. How we set $\eta_t$ and $\lambda_t$ is detailed in the Appendix.

To see how `FedEx` is approximately optimizing the relaxed objective (7), we can consider the case where $\text{Alg}_a$ is Reptile [39], which was designed to optimize some approximation of (6) for fixed $c$, or equivalently the relaxed objective for an atomic distribution $\mathcal{D}_\theta$. The theoretical literature on meta-learning [20, 21] shows that Reptile can be interpreted as optimizing a surrogate objective minimizing the squared distance between $\mathbf{w}$ and the optimum of each task $i$, with the latter being replaced by the last iterate in practice. It is also shown that the surrogate objective is useful for personalization in the online convex setting.[4] As opposed to this past work, `FedEx` makes two gradient updates in the outer loop, on two disjoint sets of variables: the first is the sub-routine $\text{Agg}_b$ of $\text{Alg}_a$ that aggregates the outputs of local training and is using the gradient of the surrogate objective, since the derivative of the squared distance is the difference between the initialization $\mathbf{w}$ and the parameter at the last iterate of $\text{Loc}_c$; the second is the exponentiated gradient update that is directly using an unbiased estimate of the derivative of the second objective w.r.t. the distribution parameters $\theta$. Thus, roughly speaking `FedEx` runs simultaneous stochastic gradient descent on the relaxed objective (7), although for the variables $\mathbf{w}$ we are using a first-order surrogate. In the theoretical portion of this work we employ this interpretation to show the approach works for tuning the step-size of online gradient descent in the online convex optimizations setting.

**Wrapping** `FedEx`    We can view `FedEx` as an algorithm of the form tuned by Algorithm 1 that implements federated training of a supernet parameter $(\mathbf{w}, \theta)$, with the local training routine $\text{Loc}$ including a step for sampling $c \sim \mathcal{D}_\theta$ and the server aggregation routine including an exponentiated update of $\theta$. Thus we can wrap `FedEx` in Algorithm 1, which we find useful for a variety of reasons:

- The wrapper can tune the settings of $b$ for the aggregation step $\text{Agg}_b$, which `FedEx` cannot.
- `FedEx` itself has a few hyperparameters, e.g. how to set the baseline $\lambda_t$, which can be tuned.

---

[3]We will use some wrapper algorithm to tune the hyperparameters $b$ of $\text{Agg}_b$.

[4]Formally they study a sequence of upper bounds and not a surrogate objective, as their focus is online learning.

**Algorithm 2:** `FedEx`

**Input:** configurations $c_1, \ldots, c_k \in \mathcal{C}$, setting $b$ for
$\quad\quad$ $\texttt{Agg}_b$, schemes for setting step-size $\eta_t$ and
$\quad\quad$ baseline $\lambda_t$, total number of steps $\tau \geq 1$
initialize $\theta_1 = \mathbf{1}_k/k$ and shared weights $\mathbf{w}_1 \in \mathbb{R}^d$
**for** *comm. round* $t = 1, \ldots, \tau$ **do**
$\quad$ **for** *client* $i = 1, \ldots, B$ **do**
$\quad\quad$ send $\mathbf{w}_t, \theta_t$ to client
$\quad\quad$ sample $c_{ti} \sim \mathcal{D}_{\theta_t}$
$\quad\quad$ $\mathbf{w}_{ti} \leftarrow \texttt{Loc}_{c_{ti}}(T_{ti}, \mathbf{w}_t)$
$\quad\quad$ send $\mathbf{w}_{ti}, c_{ti}, L_{V_{ti}}(\mathbf{w}_{ti})$ to server
$\quad$ $\mathbf{w}_{t+1} \leftarrow \texttt{Agg}_b(\mathbf{w}, \{\mathbf{w}_{ti}\}_{i=1}^B)$
$\quad$ $\tilde{\nabla}_j \leftarrow \dfrac{\sum_{i=1}^B |V_{ti}|(L_{V_{ti}}(\mathbf{w}_{ti}) - \lambda_t)\mathbf{1}_{c_{ti}=c_j}}{\theta_{t[j]} \sum_{i=1}^B |V_{ti}|} \; \forall \; j$
$\quad$ $\theta_{t+1} \leftarrow \theta_t \odot \exp(-\eta_t \tilde{\nabla})$
$\quad$ $\theta_{t+1} \leftarrow \theta_{t+1}/\|\theta_{t+1}\|_1$
**Output:** model $\mathbf{w}$, hyperparameter distribution $\theta$

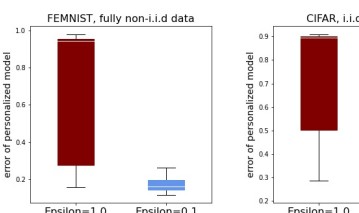 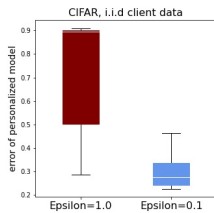

Figure 3: Comparison of the range of performance values attained using different perturbation settings. Although the range is much smaller for $\epsilon = 0.1$ than for $\epsilon = 1.0$ (the latter is the entire space), it still covers a large (roughly 10-20%) range of different performance levels on both FEMNIST (left) and CIFAR (right).

- By running multiple seeds and potentially using early stopping, we can run `FedEx` using more aggressive steps-sizes and the wrapper will discard cases where this leads to poor results.
- We can directly compare `FedEx` to a regular hyperparameter optimization scheme run over the original algorithm, e.g. `FedAvg`, by using the same scheme to both wrap `FedEx` and tune `FedAvg`.
- Using the wrapper allows us to determine the configurations $c_1, \ldots, c_k$ given to Algorithm 2 using a local perturbation scheme (detailed next) while still exploring the entire hyperparameter space.

**Local Perturbation** It remains to specify how to select the configurations $c_1, \ldots, c_k \in \mathcal{C}$ to pass to Algorithm 2. While the simplest approach is to draw from $\text{Unif}^k(\mathcal{C})$, we find that this leads to unstable behavior if the configurations are too distinct from each other. To interpolate between sampling $c_i$ independently and setting them to be identical (which would just be equivalent to the baseline algorithm), we use a simple *local perturbation* method in which $c_1$ is sampled from $\text{Unif}(\mathcal{C})$ and $c_2, \ldots, c_k$ are sampled uniformly from a local neighborhood of $\mathcal{C}$. For continuous hyperparameters (e.g. step-size, dropout) drawn from an interval $[a, b] \subset \mathbb{R}$ the local neighborhood is $[c \pm (b-a)\varepsilon]$ for some $\varepsilon \geq 0$, i.e. a scaled $\varepsilon$-ball; for discrete hyperparameters (e.g. batch-size, epochs) drawn from a set $\{a, \ldots, b\} \subset \mathbb{Z}$, the local neighborhood is similarly $\{c - \lfloor (b-a)\varepsilon \rfloor, \ldots, c + \lceil (b-a)\varepsilon \rceil\}$; in our experiments we set $\varepsilon = 0.1$, which works well, but run ablation studies varying these values in the appendix showing that a wide range of them leads to improvement. Note that while local perturbation does limit the size of the search space explored by each instance of `FedEx`, as shown in Figure 3 the difference in performance between different configurations in the same ball is still substantial.

**Limitations of** `FedEx` While `FedEx` is applicable to many important FL algorithms, those that cannot be decomposed into local fine-tuning and aggregation should instead be tuned by one of our baselines, e.g. SHA. `FedEx` is also limited in that it is forced to rely on such algorithms as wrappers for tuning its own hyperparameters and certain FL hyperparameters such as server learning rate.

## 4 Theoretical Analysis for Tuning the Step-Size in an Online Setting

As noted in Section 3, `FedEx` can be viewed as alternating minimization, with a gradient step on a surrogate personalization loss and an exponentiated gradient update of the configuration distribution $\theta$. We make this formal and prove guarantees for a simple variant of `FedEx` in the setting where the server has one client per round, to which the server sends an initialization to solve an online convex optimization (OCO) problem using online gradient descent (OGD) on a sequence of $m$ adversarial convex losses (i.e. one SGD epoch in the stochastic case). Note we use "client" and "task" interchangeably, as the goal is a meta-learning (personalization) result. The performance measure here is *task-averaged regret*, which takes the average over $\tau$ clients of the regret they incur on its loss:

$$\bar{\mathbf{R}}_\tau = \frac{1}{\tau} \sum_{t=1}^{\tau} \sum_{i=1}^{m} \ell_{t,i}(\mathbf{w}_{t,i}) - \ell_{t,i}(\mathbf{w}_t^*) \tag{8}$$

Here $\ell_{t,i}$ is the $i$th loss of client $t$, $\mathbf{w}_{t,i}$ the parameter chosen on its $i$th round from a compact parameter space $\mathcal{W}$, and $\mathbf{w}_t^* \in \arg\min_{\mathbf{w} \in \mathcal{W}} \sum_{i=1}^{m} \ell_{t,i}(\mathbf{w})$ the task optimum. In this setting, the Average Regret-Upper-Bound Analysis (ARUBA) framework [20] can be used to show guarantees for a `Reptile` (i.e. `FedEx` with a server step-size) variant in which at each round the initialization is updated as $\mathbf{w}_{t+1} \leftarrow (1 - \alpha_t)\mathbf{w}_t + \alpha_t \mathbf{w}_t^*$ for server step-size $\alpha_t = 1/t$. Observe that the only difference between this update and `FedEx`'s is that the task-$t$ optimum $\mathbf{w}_t^*$ is used rather than the last iterate of OGD on that task. Specifically they bound task-averaged regret by

$$\bar{\mathbf{R}}_\tau \leq \tilde{\mathcal{O}} \left( \frac{1}{\sqrt[4]{\tau}} + V \right) \sqrt{m} \quad \text{for} \quad V^2 = \min_{\mathbf{w} \in \mathcal{W}} \frac{1}{\tau} \sum_{t=1}^{\tau} \|\mathbf{w} - \mathbf{w}_t^*\|_2^2 \tag{9}$$

Here $V$—the average deviation of the optimal actions $\mathbf{w}_t^*$ across tasks—is a measure of *task-similarity*: $V$ is small when the tasks (clients) have similar data and thus can be solved by similar parameters in $\mathcal{W}$ but large when their data is different and so the optimum parameters to use are very different. Thus the bound in (9) shows that as the server (meta-learning) sees more and more clients (tasks), their regret on each decays with rate $1/\sqrt[4]{\tau}$ to depend only on the task-similarity, which is hopefully small if the client data is similar enough that transfer learning makes sense, in particular if $V \ll \text{diam}(\mathcal{W})$. Since single-task regret has lower bound $\Omega(D\sqrt{m})$, achieving asymptotic regret $V\sqrt{m}$ thus demonstrates successful learning of a useful initialization in $\mathcal{W}$ that can be used for personalization. Note that such bounds can also be converted to obtain guarantees in the statistical meta-learning setting as well [20].

A drawback of past results using the ARUBA framework is that they either assume the task-similarity $V$ is known in order to set the client step-size [25] or they employ an OCO method to learn the local step-size that cannot be applied to other potential algorithmic hyperparameters [20]. In contrast, we prove results for using bandit exponentiated gradient to tune the client step-size, which is precisely the `FedEx` update. In particular, Theorem 4.1 shows that by using a discretization of potential client step-sizes as the configurations in Algorithms 2 we can obtain the following task-averaged regret:

**Theorem 4.1.** *Let $\mathcal{W} \subset \mathbb{R}^d$ be convex and compact with diameter $D = \text{diam}(\mathcal{W})$ and let $\ell_{t,i}$ be a sequence of $m\tau$ $b$-bounded convex losses—$m$ for each of $\tau$ tasks—with Lipschitz constant $\leq G$. We assume that the adversary is oblivious within-task. Suppose we run Algorithm 2 with $B = 1$, configurations $c_j = \frac{D}{Gj\sqrt{m}}$ for each $j = 1, \ldots, k$ determining the local step-size of single-epoch SGD (OGD), $\mathbf{w}_{ti} = \mathbf{w}_t^*$, regret $\sum_{i=1}^{m} \ell_{t,i}(\mathbf{w}_{t,i}) - \ell_{t,i}(\mathbf{w}_t)$ used in place of $L_{V_{ti}}(\mathbf{w}_{ti})$, and $\lambda_t = 0 \,\forall\, t \in [\tau]$. Then if $\eta_t = \frac{1}{mb}\sqrt{\frac{\log k}{k\tau}} \,\forall\, t \in [\tau]$, $k^{\frac{3}{2}} = \frac{DG}{b}\sqrt{\frac{\tau}{2m}}$, and $\text{Agg}_b(\mathbf{w}, \mathbf{w}^*) = (1 - \alpha_t)\mathbf{w} + \alpha_t \mathbf{w}_t^*$ for $\alpha_t = 1/t \,\forall\, t \in [\tau]$ we have (taking expectations over sampling from $D_{\theta_t}$)*

$$\mathbb{E}\bar{\mathbf{R}}_\tau \leq \tilde{\mathcal{O}} \left( \sqrt[3]{m/\tau} + V \right) \sqrt{m} \tag{10}$$

The proof of this result, given in the supplement, follows the ARUBA framework of using meta OCO algorithm to optimize the initialization-dependent upper bound on the regret of OGD; in addition we bound errors to the bandit setting and discretization of the step-sizes. Theorem 4.1 demonstrates that `FedEx` is a sensible algorithm for tuning the step-size in the meta-learning setting where each task is an OCO problem, with the average regret across tasks (clients) converging to depend only on the task-similarity $V$, which we hope is small in the setting where personalization is useful. As we can see by comparing to the bound in (9), besides holding for a more generally-applicable algorithm our bound also improves the dependence on $\tau$, albeit at the cost of an additional $m^{\frac{1}{3}}$ factor. Note that that the sublinear term can be replaced by $1/\sqrt{\tau}$ in the full-information setting, i.e. where required the client to try SGD with each configuration $c_j$ at each round to obtain regret for all of them.

Table 1: Final test error obtained when tuning using a standard hyperparameter tuning algorithm (SHA or RS) alone, or when using it for server (aggregation) hyperparameters while `FedEx` tunes client (on-device training) hyperparameters. The target model is the one used to compute on-device validation error by the wrapper method, as well as the one used to compute test error after tuning. Note that this table reports the final error results corresponding to the online evaluations reported in Figure 4, which measure performance as more of the computational budget is expended.

| Wrapper method | Target model | Tuning method | Shakespeare i.i.d. | Shakespeare non-i.i.d. | FEMNIST i.i.d. | FEMNIST non-i.i.d. | CIFAR-10 i.i.d. |
|---|---|---|---|---|---|---|---|
| Random Search (RS) | global | RS (server & client) | $60.32 \pm 10.03$ | $64.36 \pm 14.19$ | $22.81 \pm 4.56$ | $22.98 \pm 3.41$ | $30.46 \pm 9.44$ |
| | | + FedEx (client) | $53.94 \pm 9.13$ | $57.70 \pm 17.57$ | $20.96 \pm 4.77$ | $22.30 \pm 3.66$ | $34.83 \pm 14.74$ |
| | person-alized | RS (server & client) | $61.10 \pm 9.32$ | $61.71 \pm 9.08$ | $17.45 \pm 2.82$ | $17.77 \pm 2.63$ | $34.89 \pm 10.56$ |
| | | + FedEx (client) | $54.90 \pm 9.97$ | $56.48 \pm 13.60$ | $16.31 \pm 3.77$ | $15.93 \pm 3.06$ | $39.13 \pm 15.13$ |
| Successive Halving (SHA) | global | SHA (server & client) | $47.38 \pm 3.40$ | $46.79 \pm 3.51$ | $18.64 \pm 1.68$ | $20.30 \pm 1.66$ | $21.62 \pm 2.51$ |
| | | + FedEx (client) | $\mathbf{44.52 \pm 1.68}$ | $\mathbf{45.24 \pm 3.31}$ | $19.22 \pm 2.05$ | $19.43 \pm 1.45$ | $\mathbf{20.82 \pm 1.37}$ |
| | person-alized | SHA (server & client) | $46.77 \pm 3.61$ | $48.04 \pm 3.72$ | $\mathbf{14.79 \pm 1.55}$ | $14.78 \pm 1.31$ | $24.81 \pm 6.13$ |
| | | + FedEx (client) | $46.08 \pm 2.57$ | $45.89 \pm 3.76$ | $14.97 \pm 1.31$ | $\mathbf{14.76 \pm 1.70}$ | $21.77 \pm 2.83$ |

## 5 Empirical Results

In our experiments, we instantiate `FedEx` on the problem of tuning `FedAvg`, `FedProx`, and `Reptile`; the first is the most popular algorithm for federated training, the second is an extension designed for heterogeneous devices, and the last is a compatible meta-learning method used for learning initializations for personalization. At communication round $t$ these algorithms use the aggregation

$$\texttt{Agg}_b(\mathbf{w}, \{\mathbf{w}_i\}_{i=1}^B) = (1 - \alpha_t)\mathbf{w} + \frac{\alpha_t}{\sum_{i=1}^B |T_{ti}|} \sum_{i=1}^B |T_{ti}|\mathbf{w}_i \tag{11}$$

for some learning rate $\alpha_t > 0$ that can vary through time; in the case of `FedAvg` we have $\alpha_t = 1 \, \forall \, t$. The local training sub-routine $\texttt{Loc}_c$ is SGD with hyperparameters $c$ over some objective defined by the training data $T_{ti}$, which can also depend on $c$. For example, to include `FedProx` we include in $c$ an additional local hyperparameter for the proximal term compared with that of `FedAvg`.

We tune several hyperparameters of both aggregation and local training; for the former we tune the server learning rate schedule and momentum, found to be helpful for personalization [17]; for the latter we tune the learning rate, momentum, weight-decay, the number of local epochs, the batch-size, dropout, and proximal regularization. Please see the supplementary material for the exact hyperparameter space considered. While we mainly evaluate `FedEx` in cross-device federated settings, which is generally more difficult than cross-silo in terms of hyperparameter optimization, `FedEx` can be naturally applied to cross-silo settings, where the challenges of heterogeneity, privacy requirements, and personalization remain.

Because our baseline is running Algorithm 1, a standard hyperparameter tuning algorithm, to tune all hyperparameters, and because we need to also wrap `FedEx` in such an algorithm for the reasons described in Section 3, our empirical results will test the following question: does `FedEx`, wrapped by random search (RS) or a successive halving algorithm (SHA), do better than RS or SHA run with the same settings directly? Here "better" will mean both the final test accuracy obtained and the online evaluation setting, which tests how well hyperparameter optimization is doing at intermediate phases. Furthermore, we also investigate whether `FedEx` can improve upon the wrapper alone even when targeting a good *global* and not personalized model, i.e. when elimination decisions are made using the average global validation loss. We run Algorithm 1 on the personalized objective and use RS and SHA with elimination rate $\eta = 3$, the latter following Hyperband [26]. To both wrappers we allocate the same (problem-dependent) tuning budget. To obtain the elimination rounds in Algorithm 1 for SHA, we set the number of eliminations to $R = 3$, fix a total communication round budget, and fix a maximum number of rounds to be allocated to any configuration $a$; as detailed in the Appendix, this allows us to determine $T_1, \ldots, T_R$ so as to use up as much of the budget as possible.

We evaluate the performance of `FedEx` on three datasets (Shakespeare, FEMNIST, and CIFAR-10) on both vision and language tasks. We consider the following two different partitions of data:

1. Each device holds i.i.d. data. While overall data across the entire network can be non-i.i.d., we randomly shuffle local data *within* each device before splitting into train, validation, and test sets.

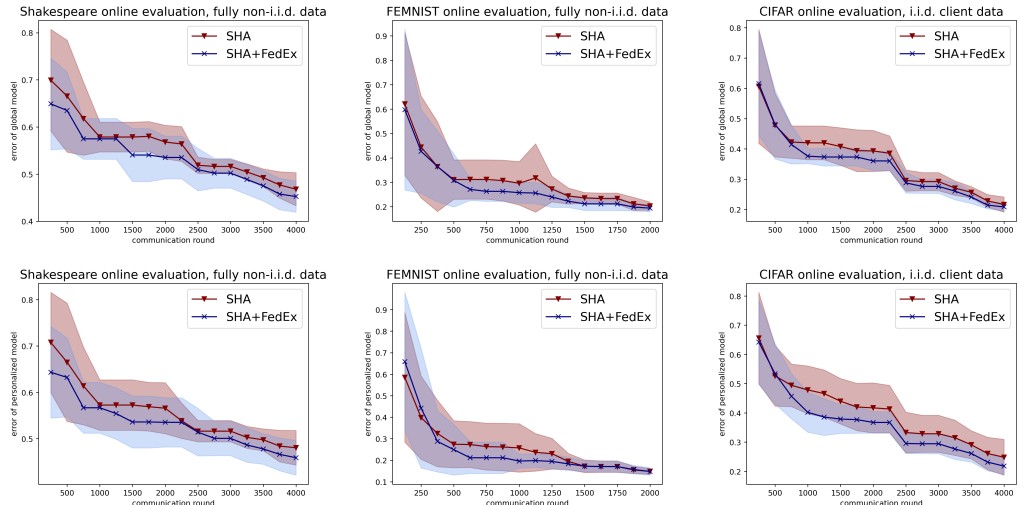

Figure 4: Online evaluation of `FedEx` on the Shakespeare next-character prediction dataset (left), the FEMNIST image classification dataset (middle), and the CIFAR-10 image classification dataset (right) in the fully non-i.i.d. setting (except CIFAR-10). We report global model performance on the top and personalized performance on the bottom. All evaluations are run for three trials.

2. Each device holds non-i.i.d. data. In Shakespeare, each device is an actor and the local data is split according to the temporal position in the play; in FEMNIST, each device is the digit writer and the local data is split randomly; in CIFAR-10, we do not consider a non-i.i.d. setting.

For Shakespeare and FEMNIST we use 80% of the data for training and 10% each for validation and testing. In CIFAR-10 we hold out 10K examples from the usual training/testing split for validation. The backbone models used for Shakespeare and CIFAR-10 follow from the `FedAvg` evaluation [36] and use 4K communications rounds (at most 800 round for each arm), while that of FEMNIST follows from LEAF [5] and uses 2K communication rounds (at most 200 for each arm).

Table 1 presents our main results, displaying the final test error of the target model after tuning using either a wrapper algorithm alone or its combination with `FedEx`. The evaluation shows that using `FedEx` on the client parameters is either equally or more effective in most cases; in particular, a `FedEx`-modified method performs best everywhere except i.i.d. FEMNIST, where it is very close. Furthermore, `FedEx` frequently improves upon the wrapper algorithm by 2 or more percentage points.

We further present online evaluation results in Figure 4, where we display the test error of `FedEx` wrapped with SHA compared to SHA alone as a function of communication rounds. Here we see that for most of training `FedEx` is either around the same or better then the alternative, except at the beginning; the former is to be expected since the randomness of `FedEx` leads to less certain updates at initialization. Nevertheless `FedEx` is usually better than the SHA baseline by the halfway point.

## 6 Conclusion

In this paper we study the problem of hyperparameter optimization in FL, starting with identifying the key challenges and proposing reasonable baselines that adapts standard approaches to the federated setting. We further make a novel connection to the weight-sharing paradigm from NAS—to our knowledge the first instance of this being used for regular (non-architectural) hyperparameters—and use it to introduce `FedEx`. This simple, low-overhead algorithm for accelerating the tuning of hyperparameters in federated learning can be theoretically shown to successfully tune the step-size for multi-task OCO problems and effectively tunes `FedAvg`, `FedProx`, and `Reptile` on standard benchmarks. The scope of application of `FedEx` is very broad, including tuning actual architectural hyperparameters rather than just settings of local SGD, i.e. doing federated NAS, and tuning initialization-based meta-learning algorithms such as `Reptile` and `MAML`. Lastly, any work on FL comes with privacy and fairness risks due its frequent use of sensitive data; thus any application of our work must consider tools being developed by the community for mitigating such issues [32, 37].

## Acknowledgments

This material is based on work supported by the National Science Foundation under grants CCF-1535967, CCF-1910321, IIS-1618714, IIS-1901403, SES-1919453, IIS-1705121, IIS-1838017, IIS-2046613 and IIS-2112471; the Defense Advanced Research Projects Agency under cooperative agreements HR00112020003 and FA875017C0141; an AWS Machine Learning Research Award; an Amazon Research Award; a Bloomberg Research Grant; a Microsoft Research Faculty Fellowship; an Amazon Web Services Award; a Facebook Faculty Research Award; funding from Booz Allen Hamilton Inc.; a Block Center Grant; and a Two Sigma Fellowship Award. Any opinions, findings and conclusions or recommendations expressed in this material are those of the author(s) and do not necessarily reflect the views of any of these funding agencies.

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
