# A   Proof of Theorem 4.1

*Proof.* Let $\gamma_t \sim \mathcal{D}_{\theta_t}$ be the step-size chosen at time $t$. Then we have that

$$
\begin{aligned}
\tau \mathbb{E}\bar{\mathbf{R}}_\tau &= \sum_{t=1}^{\tau} \mathbb{E}_{\gamma_t} \sum_{i=1}^{m} \ell_{t,i}\left(\mathbf{w}_{t,i}^{(\mathbf{w}_t,\gamma_t)}\right) - \sum_{i=1}^{m} \ell_{t,i}(\mathbf{w}_t^*) \\
&= \sum_{t=1}^{\tau} \sum_{j=1}^{k} \theta_{t[j]} \sum_{i=1}^{m} \ell_{t,i}\left(\mathbf{w}_{t,i}^{(\mathbf{w}_t,c_j)}\right) - \sum_{i=1}^{m} \ell_{t,i}(\mathbf{w}_t^*) \\
&\leq \frac{\log k}{\eta} + \eta k \tau m^2 b^2 \\
&\quad + \min_{j\in[k]} \sum_{t=1}^{\tau} \sum_{i=1}^{m} \ell_{t,i}\left(\mathbf{w}_{t,i}^{(\mathbf{w}_t,c_j)}\right) - \min_{\mathbf{w}\in\mathcal{W}} \sum_{i=1}^{m} \ell_{t,i}(\mathbf{w}_t^*) \\
&\leq 2mb\sqrt{\tau k \log k} + \min_{j\in[k]} \sum_{t=1}^{\tau} \frac{1}{2c_j}\|\mathbf{w}_t - \mathbf{w}_t^*\|_2^2 + c_j m G^2 \\
&\leq 2mb\sqrt{\tau k \log k} + \min_{j\in[k]} \frac{D^2(1+\log\tau)}{2c_j} + \left(\frac{V^2}{2c_j} + c_j m G^2\right)\tau \\
&\leq 2mb\sqrt{\tau k \log k} + \frac{D^2(1+\log\tau) + V^2\tau}{2\gamma^*} + \gamma^* m G^2 \tau \\
&\quad + \min_{j\in[k]} \left(\frac{1}{c_j} - \frac{1}{\gamma^*}\right) \frac{D^2(1+\log\tau) + V^2\tau}{2} + (c_j - \gamma^*)m G^2 \tau \\
&\leq 2mb\sqrt{\tau k \log k} + 4D\sqrt{\frac{\tau + \tau\log\tau}{2}} + \left(2V + \frac{D}{k}\right)G\tau\sqrt{\frac{m}{2}} \\
&= mb\sqrt{2\tau\log\tau} + 4D\sqrt{\frac{\tau + \tau\log\tau}{2}} + (DG + 2GV\tau)\sqrt{\frac{m}{2}}
\end{aligned}
$$

where the second line uses linearity of expectations over $\gamma_t \sim \mathcal{D}_{\theta_t}$, the third substitutes the bandit regret of EG [? , Corollary 4.2], the fourth substitutes $\eta = \frac{1}{mb}\sqrt{\frac{\log k}{\tau k}}$ and the regret of OGD [? , Corollary 2.7], the fifth substitutes the regret guarantee of Adaptive OGD over functions $\frac{1}{2}\|\mathbf{w}_t - \mathbf{w}_t^*\|_2^2$ [? , Theorem 2.1] with step-size $\alpha_t = 1/t$ and the definition of $V$, the sixth substitutes the best discretized step-size $c_j$ for the optimal $\gamma^* \in \left(0, \frac{D}{G\sqrt{2m}}\right]$, and the seventh substitutes $\frac{V}{2G\sqrt{2m}} + \frac{D}{2G}\sqrt{\frac{1+\log\tau}{2m\tau}}$ for $\gamma^*$ and $\arg\min_{j:c_j\geq\gamma^*}$ for $\arg\min_j c_j$. Setting $k^{\frac{3}{2}} = \frac{DG}{b}\sqrt{\frac{\tau}{2m}}$ and dividing both sides by $\tau$ yields the result. $\qquad\square$

# B  Decomposing Federated Optimization Methods

As detailed in Section 2 our analysis and use of `FedEx` to tune local training hyperparameters depends on a formulation that decomposes FL methods into two subroutines: a local training routine $\text{Loc}_c(S, \mathbf{w})$ with hyperparameters $c$ over data $S$ and starting from initialization $\mathbf{w}$ and an aggregation routine $\text{Agg}_b$ with hyperparameters $b$. In this section we discuss how a variety of federated optimization methods, including several of the best-known, can be decomposed in this manner. This enables the application of `FedEx` to tune their hyperparameters.

## B.1  `FedAvg` [36]

The best-known FL method, `FedAvg` runs SGD on each client in a batch starting from a shared initialization and then updates to the average of the last iterate of the clients, often weighted by the number of data points each client has. The decomposition here is:

$\text{Loc}_c$  Local SGD (or another gradient-based algorithm, e.g. Adam [? ]), with $c$ being the standard hyperparameters such as step-size, momentum, weight-decay, etc.

$\text{Agg}_b$  Weighted averaging, with no hyperparameters in $b$.

## B.2  `FedProx` [31]

`FedProx` has the same decomposition as `FedAvg` except local SGD is replaced by a proximal version that regularizes the routine to be closer to the initialization, adding another hyperparameter to $c$ governing the strength of this regularization.

## B.3  `Reptile` [39]

A well-known meta-learning algorithm, `Reptile` has the same decomposition as `FedAvg` except the averaged aggregation is replaced by a convex combination of the initialization and the average of the last iterates, as in Equation 11. This adds another hyperparameter to $b$ governing the tradeoff between the two.

## B.4  `SCAFFOLD` [19]

`SCAFFOLD` comes in two variants, both of which compute and aggregate control variates in parallel to the model weights. The decomposition here is:

$\text{Loc}_c$  Local SGD starting from a weight initialization with a control variate, which can be merged to form the local training initialization. The hyperparameters in $c$ are the same as in `FedAvg`.

$\text{Agg}_b$  Weighted averaging of both the initialization and the control variates, with the same hyperparameters as `Reptile`.

## B.5  `FedDyn` [1]

In addition to a `FedAvg`/`FedProx`/`Reptile`-like training routine, this algorithm maintains a regularizer on each device that affects the local training routine. While this statefulness cannot strictly be subsumed in our decomposition, since it does not introduce any additional hyperparameters the remaining hyperparameters can be tuned in the same manner as we do for `FedAvg`/`FedProx`/`Reptile`. In order to choose between using `FedDyn` or not, one can introduce a binary hyperparameter to $c$ specifying whether or not $\text{Loc}_c$ uses that term in the objective it optimizes or not, allowing it also to be tuned via `FedEx`.

## B.6  `FedPA` [2]

This algorithm replaces local SGD in `FedAvg` by a local Markov-chain Monte Carlo (MCMC) routine starting from the initialization given by aggregating the previous round's MCMC routines. The decomposition is then just a replacement of local SGD and its associated hyperparameters by local MCMC and its hyperparameters, with the aggregation routine remaining the same.

## B.7 `Ditto` [29]

Although it depends on what solver is used for the local solver and aggregation routines, in the simplest formulation, the local optimization of personalized models involves an additional regularization hyperparameter. While the updating rule of `Ditto` is different from that of `FedProx`, the hyperparameters can be decomposed and tuned in a similar manner.

## B.8 `MAML` [? ]

A well-known meta-learning algorithm, `MAML` takes one or more full-batch gradient descent (GD) steps locally and updates the global model using a second-order gradient using validation data. The decomposition here is :

$\text{Loc}_c$ Local SGD starting from a weight initialization. The hyperparameters in $c$ are the same as in `FedAvg`. The algorithm also returns second-order information required to compute the meta-gradient.

$\text{Agg}_b$ Meta-gradient computation, summation, and updating using a standard descent method like Adam [? ]. The hyperparameters in $b$ are the hyperparameters of the latter.

# C  `FedEx` Details

## C.1  Stochastic Gradient used by `FedEx`

Below is a simple calculation showing that the stochastic gradient used to update the categorical architecture distribution of `FedEx` is an unbiased approximation of the true gradient w.r.t. its parameters.

$$
\begin{aligned}
\nabla_{\theta_j} & \mathbb{E}_{c_{ij}|\theta} L_{V_{ti}}(\mathbf{w}_i) \\
&= \nabla_{\theta_j} \mathbb{E}_{c_{ij}|\theta} (L_{V_{ti}}(\mathbf{w}_i) - \lambda) \\
&= \mathbb{E}_{c_{ij}|\theta} \left( (L_{V_{ti}}(\mathbf{w}_i) - \lambda) \nabla_{\theta_j} \log \mathbb{P}_\theta(c_{ij}) \right) \\
&= \mathbb{E}_{c_{ij}|\theta} \left( (L_{V_{ti}}(\hat{w}_k) - \lambda) \nabla_{\theta_j} \log \prod_{i=1}^{n} \mathbb{P}_\theta(c_{ij} = c_j) \right) \\
&= \mathbb{E}_{c_{ij}|\theta} \left( (L_{V_{ti}}(\mathbf{w}_i) - \lambda) \sum_{i=1}^{n} \nabla_{\theta_j} \log \mathbb{P}_\theta(c_{ij} = c_j) \right) \\
&= \mathbb{E}_{c_{ij}|\theta} \left( \frac{(L_{V_{ti}}(\mathbf{w}_i) - \lambda) 1_{c_{ij}=c_j}}{\theta_j} \right)
\end{aligned}
$$

Note that this use of the reparameterization trick has some similarity with a recent RL approach to tune the local step-size and number of epochs [38]; however, `FedEx` can be rigorously formulated as an optimization over the personalization objective, has provable guarantees in a simple setting, uses a different configuration distribution that leads to our exponentiated update, and crucially for practical deployment does not depend on obtaining aggregate reward signal on each round.

## C.2 `FedEx` **wrapped with SHA**

For completeness, we present the pseudo code of wrapping `FedEx` with SHA in Algorithm 3 below.

---

**Algorithm 3:** `FedEx` wrapped with SHA

---

**Input:** distribution $\mathcal{D}$ over hyperparameters $\mathcal{A}$, elimination rate $\eta \in \mathbb{N}$, elimination rounds
$\quad\quad \tau_0 = 0, \tau_1, \ldots, \tau_R$
sample set of $\eta^R$ hyperparameters $H \sim \mathcal{D}^{[\eta^R]}$
initialize a model $\mathbf{w}_a \in \mathbb{R}^d$ for each $a \in H$
**for** *elimination round $r \in [R]$* **do**
$\quad$ **for** *setting $a = (b, c) \in H$* **do**
$\quad\quad \lfloor\ s_a, \mathbf{w}_a, \theta_a \leftarrow \texttt{FedEx}\,(\mathbf{w}_a, b, c, \theta_a, \tau_{r+1} - \tau_r)$
$\quad\ H \leftarrow \{a \in H : s_a \leq \frac{1}{\eta}\text{-quantile}(\{s_a : a \in H\})\}$
**Output:** remaining $a \in H$ and associated model $\mathbf{w}_a$

---

$\texttt{FedEx}\,(\mathbf{w}, b, \{c_1, \ldots, c_k\}, \theta, \tau \geq 1):$

---

initialize $\theta_1 \leftarrow \theta$
initialize shared weights $\mathbf{w}_1 \leftarrow \mathbf{w}$
**for** *comm. round $t = 1, \ldots, \tau$* **do**
$\quad$ **for** *client $i = 1, \ldots, B$* **do**
$\quad\quad$ send $\mathbf{w}_t, \theta_t$ to client
$\quad\quad$ sample $c_{ti} \sim \mathcal{D}_{\theta_t}$
$\quad\quad \mathbf{w}_{ti} \leftarrow \texttt{Loc}_{c_{ti}}(T_{ti}, \mathbf{w}_t)$
$\quad\quad$ send $\mathbf{w}_{ti}, c_{ti}, L_{V_{ti}}(\mathbf{w}_{ti})$ to server
$\quad\ \mathbf{w}_{t+1} \leftarrow \texttt{Agg}_b(\mathbf{w}, \{\mathbf{w}_{ti}\}_{i=1}^B)$
$\quad\ $ set step size $\eta_t$ and baseline $\lambda_t$
$\quad\ \tilde{\nabla}_j \leftarrow \frac{\sum_{i=1}^B |V_{ti}|(L_{V_{ti}}(\mathbf{w}_{ti}) - \lambda_t)1_{c_{ti} = c_j}}{\theta_{t[j]} \sum_{i=1}^B |V_{ti}|}\ \forall\ j$
$\quad\ \theta_{t+1} \leftarrow \theta_t \odot \exp(-\eta_t \tilde{\nabla})$
$\quad\ \theta_{t+1} \leftarrow \theta_{t+1}/\|\theta_{t+1}\|_1$
$\quad\ s \leftarrow \sum_{i=1}^B |V_{ti}| L_{V_{ti}} / \sum_{i=1}^B |V_{ti}|$
$\quad$ **Return** $s$, model $\mathbf{w}$, hyperparameter distribution $\theta$

---

## C.3 Hyperparameters of `FedEx`

We tune the computation of the baseline $\lambda_t$, which we set to

$$\lambda_t = \frac{1}{\sum_{s<t} \gamma^{t-s}} \sum_{s<t} \frac{\gamma^{t-s}}{\sum_{i=1}^B |V_{ti}|} \sum_{i=1}^B L_{V_{ti}}(\mathbf{w}_i)$$

for discount factor $\gamma \in [0, 1]$. As discussed in Section 3, the local perturbation factor is set to $\varepsilon = 0.1$. 27 configurations are used in each arm for SHA and RS. The number of configuration used per arm of `FedEx` (i.e. the dimensionality of $\theta$) is the same (27).

# D   Experimental Details

Code implementing `FedEx` is available at `https://github.com/mkhodak/fedex`. The code automatically downloads CIFAR-10 data, while Shakespeare and FEMNIST data is made available by the LEAF repository: `https://github.com/TalwalkarLab/leaf`.

## D.1   Settings of the Baseline/Wrapper Algorithm

We use the same settings of Algorithm 1 for both tuning `FedAvg` and for wrapping `FedEx`. Given an elimination rate $\eta$, number of elimination rounds $R$, resource budget $B$, and maximum rounds per arm $M$, we assign $T_1, \ldots, T_R$ s.t.

$$T_i - T_{i-1} = \frac{T - M}{\frac{\eta^{n+1}-1}{\eta-1} - n - 1}$$

(recall $T_0 = 0$) and assign any remaining resources to maximize resource use. All remaining details were noted in Section 5.

## D.2   Hyperparameters of `FedAvg`/`FedProx`/`Reptile`

Server hyperparameters (learning rate $\alpha_t = \gamma^t$):

$$\begin{aligned}
\log_{10} \text{lr} : \quad & \text{Unif}[-1, 1] \\
\text{momentum} : \quad & \text{Unif}[0, 0.9] \\
\log_{10}(1 - \gamma) : \quad & \text{Unif}[-4, -2]
\end{aligned}$$

Local training hyperparameters (note we only use 1 epoch for Shakespeare to conserve computation):

$$\begin{aligned}
\log_{10}(\text{lr}) : \quad & \text{Unif}[-4, 0] \\
\text{momentum} : \quad & \text{Unif}[0.0, 1.0] \\
\log_{10}(\text{weight-decay}) : \quad & \text{Unif}[-5, -1] \\
\text{epoch} : \quad & \text{Unif}\{1, 2, 3, 4, 5\} \\
\log_2(\text{batch}) : \quad & \text{Unif}\{3, 4, 5, 6, 7\} \\
\text{dropout} : \quad & \text{Unif}[0, 0.5]
\end{aligned}$$

# E   Confidence Intervals

Table 2: Final test error obtained when tuning using a standard hyperparameter tuning algorithm (SHA or RS) alone, or when using it for server (aggregation) hyperparameters while `FedEx` tunes client (on-device training) hyperparameters. The target model is the one used to compute on-device validation error by the wrapper method, as well as the one used to compute test error after tuning. The confidence intervals displayed are 90% Student-t confidence intervals for the mean estimates from Table 1, with 5 independent trials for Shakespeare, 10 for FEMNIST, 10 for RS on CIFAR, and 6 for SHA on CIFAR.

| Wrapper method | Target model | Tuning method | Shakespeare i.i.d. | Shakespeare non-i.i.d. | FEMNIST i.i.d. | FEMNIST non-i.i.d. | CIFAR-10 i.i.d. |
|---|---|---|---|---|---|---|---|
| Random Search (RS) | global | RS (server & client) | $60.32 \pm 9.56$ | $64.36 \pm 13.53$ | $22.81 \pm 2.64$ | $22.98 \pm 1.98$ | $30.46 \pm 5.47$ |
| | | + FedEx (client) | $53.94 \pm 8.70$ | $57.70 \pm 16.75$ | $20.96 \pm 2.77$ | $22.30 \pm 2.12$ | $34.83 \pm 8.54$ |
| | personalized | RS (server & client) | $61.10 \pm 8.89$ | $61.71 \pm 8.66$ | $17.45 \pm 1.63$ | $17.77 \pm 1.52$ | $34.89 \pm 6.12$ |
| | | + FedEx (client) | $54.90 \pm 9.50$ | $56.48 \pm 12.97$ | $16.31 \pm 2.19$ | $15.93 \pm 1.77$ | $39.13 \pm 8.77$ |
| Successive Halving (SHA) | global | SHA (server & client) | $47.38 \pm 3.24$ | $46.79 \pm 3.35$ | $18.64 \pm 0.97$ | $20.30 \pm 0.96$ | $21.62 \pm 1.45$ |
| | | + FedEx (client) | $\mathbf{44.52 \pm 1.60}$ | $\mathbf{45.24 \pm 3.16}$ | $19.22 \pm 1.19$ | $19.43 \pm 0.84$ | $\mathbf{20.82 \pm 0.79}$ |
| | personalized | SHA (server & client) | $46.77 \pm 3.44$ | $48.04 \pm 3.54$ | $\mathbf{14.79 \pm 0.90}$ | $14.78 \pm 0.75$ | $24.81 \pm 3.55$ |
| | | + FedEx (client) | $46.08 \pm 2.45$ | $45.89 \pm 3.58$ | $14.97 \pm 0.76$ | $\mathbf{14.76 \pm 0.99}$ | $21.77 \pm 1.64$ |

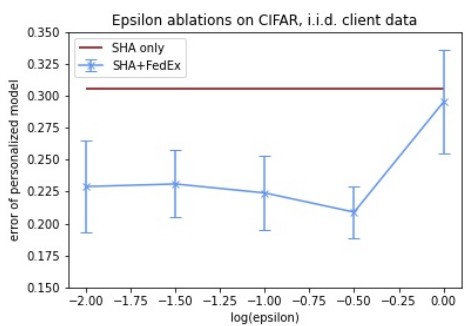

Figure 5: Comparison of different $\varepsilon$ settings for the local perturbation component of `FedEx` from Section 3.

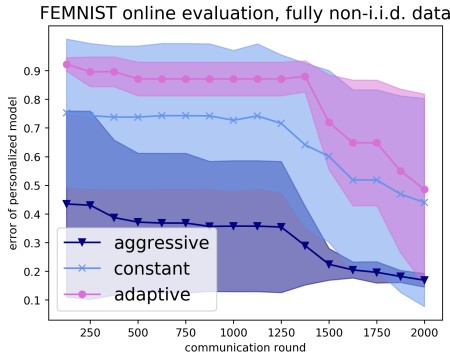

Figure 6: Comparison of step-size schedules for $\eta_t$ in `FedEx`. In practice we chose the 'aggressive' schedule, which exhibits faster convergence to favorable configurations.

## F  Ablation Studies

We now discuss two design choices of `FedEx` and how they affect performance of the algorithm. First, the choice of the local perturbation $\varepsilon = 0.1$ discussed in Section 3; we choose this setting due to its consistent performance across several settings. In Figure 5 we plot the performance of `FedEx` on CIFAR-10 between $\varepsilon = 0.0$ (no `FedEx`, i.e. SHA only) and $\varepsilon = 1.0$ (full `FedEx`, i.e. client configurations are chosen independently) and show that while the use of a nonzero $\varepsilon$ is important, performance at fairly low values of $\varepsilon$ is roughly similar.

We further investigated the setting of the step-size $\eta_t$ for the exponentiated gradient update in `FedEx`. We examine three different approaches: a constant rate of $\eta_t = \sqrt{2 \log k}$, an 'adaptive' schedule of $\eta_t = \sqrt{2 \log k}/\sqrt{\sum_{s \le t} \|\tilde{\nabla}_s\|_\infty^2}$, and an 'aggressive' schedule of $\eta_t = \sqrt{2 \log k}/\|\tilde{\nabla}_t\|_\infty$. Here $\tilde{\nabla}_t$ is the stochastic gradient w.r.t. $\theta$ computed in Algorithm 2 at step $t$ and the form of the step-size is derived from standard settings for exponentiated gradient in online learning [**?** ]. We found that the 'aggressive' schedule works best in practice, as shown in Figure 6. A key issue with using the 'constant' and 'adaptive' approaches is that they continue to assign high probability to several configurations late in the tuning process; this slows down training of the shared weights. One could consider a tradeoff between allowing `FedEx` to run longer than while keeping the total budget constant, but for simplicity we chose the more effective 'aggressive' schedule.