# OpenReview forum: "Federated Hyperparameter Tuning: Challenges, Baselines, and Connections to Weight-Sharing"
_NeurIPS.cc/2021/Conference — NeurIPS 2021 Poster_

### Official Review · Reviewer_c2J3 · 2021-07-16

**Rating:** 7
**Confidence:** 4

**Summary:**

This paper proposes a general hyperparameter tuning framework for federated learning based on the weight-sharing technique. In contrast to tuning architectural parameters as in NAS, this paper focus on tuning non-architectural parameters. The proposed framework is general enough to be applied to many FL algorithms including FedAvg, FedProx, and SCAFFOLD. A theoretical study is offered for tuning client step-size. Empirical experiments on Shakespeare, FEMNIST, and CIFAR-10 show promising results.

**Main Review:**

**Originality**: the paper presents an interesting formulation for hyperparameter tuning in federated learning. Although weight-sharing is a well-known technique in NAS for tuning architectural parameters, the formulation of adopting weight-sharing for tuning local SGD parameters in FL is an interesting idea.

**Quality**: the paper is technically sound. It gives a clean mathematical formulation with thorough explanation of motivations.

**Clarity**: the paper is well written and easy to follow.

Minor presentation issues:
- Algorithm 1, on page 4, is not introduced in the text of section 2, while being referred to a few times starting page 6.
- Equation 4, RHS misses min variable $w$ for $w \in \mathbb{R}^d$.
- Title and axis labels in all plots can be bigger for better readability.

**Strengths**

1. The proposed FedEx method is a simple and clean framework that integrates well with the local-optimization + aggregation style algorithms, such as FedAvg, FedProx, Reptile.

1. The method respects the resource limitation on federated clients and does not incur excessive computation cost. Only sampling a configuration, the client side does not have much increased workload; whereas on the sever side takes  update steps with exponentiated gradients.

1. By using weight-sharing technique and sampling from the hyperparameter distribution, the proposed method does not suffer from premature stopping from noisy elimination decisions as in SHA.

**Weaknesses**

1. Although the authors in Eq 1 and 2 present a rounded formulation on optimizing hyperparameters $a = (b, c) \in \mathcal{B} \times \mathcal{C}$, by looking at algorithm 2, FedEx can only be applied to optimizing local SGD parameters $c$ while not aggregation parameters $b$. The authors state in L214 that the wrapper in Algorithm 1 can help turning $b$, but wouldn't that mean the tuning of $b$ may suffer from issues outlined in page 4 (such as premature early stopping)?

1. The tuning scheme relies on using personalized validation error from individual clients. However, looking at Table 1, the global models obtained after aggregation almost always have lower error than personalized models for all datasets except FEMNIST, regardless of RS or SHA wrapper. Can authors offer some insights?

**Time Spent Reviewing:**

3

---

> ### Author Response · Authors · 2021-08-10
> **Response to Reviewer c2J3**
>
> Thank you for your positive review; we hope to address your questions below. Thank you also for the notes on the presentation, which we will fix in our revision.
> ### Responses to questions
> 1\. [*Although the authors in Eq 1 and 2 present a rounded formulation on optimizing hyperparameters a=(b,c)∈B×C, by looking at algorithm 2, FedEx can only be applied to optimizing local SGD parameters c while not aggregation parameters b. The authors state in L214 that the wrapper in Algorithm 1 can help turning b, but wouldn't that mean the tuning of b may suffer from issues outlined in page 4 (such as premature early stopping)?*]
> - Yes, because FedEx does not handle the aggregating parameters the benefit of using it does not extend to those parameters. However, we believe most hyperparameters in FL---including six of the nine we consider and any architectural hyperparameters in future work---are of the client algorithm. In such cases the benefits of FedEx will extend to most hyperparameters.
>
> 2\. [*The tuning scheme relies on using personalized validation error from individual clients. However, looking at Table 1, the global models obtained after aggregation almost always have lower error than personalized models for all datasets except FEMNIST, regardless of RS or SHA wrapper. Can authors offer some insights?*]
> - For CIFAR the personalized model performing worse is not very surprising because we only consider i.i.d. data there, so such a model would overfit to a few points from the distribution when the global parameter is already trained on the right distribution. For Shakespeare we believe the difficulty in the non-i.i.d. setting may be due to overfitting to the many devices with very limited data. In-practice, since the global validation loss is at least as easy to obtain as the personalized validation loss, we believe a practitioner would pick to not personalize if seeing such overfitting occurring. However, we report the results for completeness here.

---

### Official Review · Reviewer_2myh · 2021-07-16

**Rating:** 6
**Confidence:** 3

**Summary:**

The paper analyzes three challenges in hyperparameters tuning under the federated learning setting. The paper proposed two straightforward solutions (random search and successive halving) as baselines and one method, namely FedEx, as the solution to tackle those challenges. The two baselines tune hyperparameters directly. FedEx tunes hyperparameters via learning the relationship between validation scores and the hidden variables of hyperparameters. The proposed method was experimentally validated by tuning regular hyperparameters on three datasets.


**Limitations And Societal Impact:**

I have a few questions about the paper.

1. How many GPU hours does it take to reproduce the experiment?
2. The argument that "personalized losses require several forward-backward passes"(line 130 to line 135) is a little bit confusing.
3. What does this mean, and how does FedEx tackle the problem? Any example that personalized FL uses several forward-backward passes?
4. In algorithm 2, what's the meaning of "1 ctj=cj" (the Fourth line from last)
5. What’s the meaning of the shadow in figure 4. Std for multiple runs or std for clients?
6. Is Dθt (algorithm 2) different across different clients?
7. How to obtain the personalized model by FedEx?
8. The FedEx is tested with FedAvg, FedProx, and Reptile. But there is only one set of results.

**Main Review:**

Originality:
The targeting problem is to tune hyperparameters for machine learning models in federated settings. The proposed method FedEx is an interesting idea to build links between weight-sharing mechanisms and hyperparameter tuning. This mitigation allows a simultaneous searching for multiple hyperparameters in a very efficient way.

Quality:
The quality of the paper could be improved. The paper title is "Federated Hyperparameter Tuning." However, the paper only targets tuning non-architectural parameters. For non-architectural parameters tuning, the paper has a good experiment and ablation study to support the claims.

Clarity: The paper is clearly written and well organized. Details of proof and experiments are given in the supplement.

Significance: The result is important in a practical setting. Despite the paper only testing the idea with regular hyperparameter tuning. Using it on architecture search would have a promising future.

=== Post Rebuttal ===

The paper is well written, and its claims are well supported by the experiments. The connection between hyperparameter tunning and weight-sharing mechanism is an interesting point, however, it also makes the paper looks like an ensemble of two pieces.

**Time Spent Reviewing:**

6

---

> ### Author Response · Authors · 2021-08-10
> **Response to Reviewer 2myh**
>
> Thank you for your positive review; we hope to address your questions below.
> ### Responses to enumerated questions
> 1\. [*How many GPU hours does it take to reproduce the experiment?*]
> - On an NVIDIA V100: 24 for each trial of Shakespeare, 12 for each trial of CIFAR, and 2 for each trial of FEMNIST.
>
> 2\. [*The argument that "personalized losses require several forward-backward passes"(line 130 to line 135) is a little bit confusing. What does this mean, and how does FedEx tackle the problem? Any example that personalized FL uses several forward-backward passes?*]
> - This is simply stating that while computing a loss where each client uses the same (global) parameter requires just a prediction on each data-point (1 forward pass), computing a loss where each client uses a different (personalized) parameter requires first computing that parameter, i.e. training on multiple data-points and taking derivatives (so multiple forward and backward passes) if using a gradient-based personalization algorithm. In FedEx, this loss is computed while training, since the training algorithm is equivalent to personalization, and so does not need to be recomputed for evaluation purposes.
>
> 4\. [*In algorithm 2, what's the meaning of "1 ctj=cj"*]
> - This is an indicator that is one if ctj=cj and zero otherwise.
>
> 5\. [*What’s the meaning of the shadow in figure 4. Std for multiple runs or std for clients?*]
> - Std for multiple runs; we will clarify this in our revision.
>
> 6\. [*Is Dθt (algorithm 2) different across different clients?*]
> - It is the same for each client in a communication round.
>
> 7\. [*How to obtain the personalized model by FedEx?*]
> - FedEx returns an initialization parameter w and a distribution over configurations of personalization routines. To obtain the personalized model, sample a configuration from this distribution and run the personalization routine from w using that configuration to get a personalized parameter.
>
> 8\. [*The FedEx is tested with FedAvg, FedProx, and Reptile. But there is only one set of results.*]
> - FedEx automatically selects between them by including the hyperparameters associated with them as part of those being tuned.
>
> ### Response to remaining comment
> [*The quality of the paper could be improved. The paper title is "Federated Hyperparameter Tuning." However, the paper only targets tuning non-architectural parameters. For non-architectural parameters tuning, the paper has a good experiment and ablation study to support the claims.*]
> - We agree that using FedEx to accelerate the tuning of architectural hyperparameters is a natural setting for future study. We largely focus on the most commonly tuned hyperparameters---mainly non-architectural ones such as step-size, regularization, etc.---and demonstrate the connection to NAS in a simpler setting. However, it is also straightforward to apply FedEx to architecture tuning.

---

> > ### Comment · Reviewer_2myh · 2021-09-03
> > **Keep my orginal rating.**
> >
> > Thanks for the clarifications. After reading all reviewers' comments and the authors' responses, I decided to keep my original rating.

---

### Official Review · Reviewer_onB2 · 2021-07-22

**Rating:** 6
**Confidence:** 4

**Summary:**

This paper proposes a method for learning client-side hyperparameters in the federated setting by parameterizing the distribution of these hyperparameters and estimating gradients with respect to this parametererization. This approach is connected to weight-sharing methods in neural architecture search.

This paper provides a theoretical statement in a simplified setting justifying the approach taken--showing that for some settings this approach can improve average regret.

This paper also provides a reasonably thorough empirical evaluation of this approach, along with standalone implementations in the supplementary material.

**Limitations And Societal Impact:**

Societal impact is a potential strength of this paper, which could be highlighted further: the methods this paper proposes are eminently composable with standard privacy-preserving techniques in FL (like differential privacy and secure aggregation), whereas this is not necessarily the case for every imaginable approach to automating hyperparameter tuning in the federated setting.

**Main Review:**

Some fine-grained notes:

1. I find the formulation (1) a little strange--that is, there seems to be a suppressed dependence on the parameter w, or a suppressed integration over time. The output of Alg_a depends not just on T_j but also on w--so without express dependence here (or something more explicitly e.g. variational in nature) I think the formulation (1) is somewhat confusing.

2. Is there a silent assumption in SGD_c that we are actually doing SGD on something? The name gives me this intuition, but I can’t spot this in the paper itself. If not, maybe consider a rename (e.g. to Loc_c for ‘local update parameterized by c’ or something).

3. There is an additional and somewhat subtle difficulty in evaluating hyperparameters in FL which could profitably be further highlighted in the challenges and baselines section. This is the *time*-dependence of evaluation metrics in many real world FL applications. There isn't a tremendous amount of literature on this phenomenon, but one paper I am aware of is here: https://arxiv.org/pdf/1904.10120.pdf.

4. Nit: it seems like Alg1 and Fig 2 dont quite match--that is, it seems to me that Alg 1 corresponds to ‘last only’ and cant capture the other discount factors? This mismatch makes a quick read a little more difficult.

5. Another nit: a further natural baseline here (one that actually does seem relevant to me): how does performance scale here with respect to clients per round? This is relevant because of the calling-out of unnaturalness of only using last-round validation data--this becomes more reasonable as round size blows up. This would make the baselines much more compelling to me, to have a dataset of essentially unbounded scale (e.g. StackOverflow next-word-prediction) on which very large-scale training was analyzed.

6. (this is potentially genuine confusion, as I am no expert in the NAS literature) It seems to me that the connection to NAS is a little unnatural. In my mind it seems easier to consider (6) and (7) direct from (2). So it seems like NAS and weight sharing...is only there to say ‘introducing this sampling/reparameterization procedure is not a crazy thing to do’? I think this is a reasonable framing, but I'm not sure it fits with the rest of the positioning of the paper--the rest of the paper positions a little too front-and-center if my understanding is correct.

7. Nit typo (L269): s/shws/shows

8. The error bars for these table entries are *really* large. I for sure appreciate their inclusion! Checking my understanding here: these error bars represent more or less a single standard deviation around the estimated mean? IE, this std dev represents the uncertainty in the estimate of the mean itself? Assuming this is true, it seems to me that I can't really in good faith consider virtually any of these empirics to be informative. There is an alternative possibility here: that these std dev numbers rather represent the empirical standard deviation of the e.g. accuracy estimates (so that they can't be easily reduced by collecting more samples from the accuracy distribution).

This comment began as an ask for explanation of the behavior of shakespeare under random search--thinking about this is what led to closer scrutiny of the associated error bars, and made me think that this difference in point estimates may have simply been random fluctuation. This got me to thinking more deeply about the meaning of these error bars--and this thinking became a crux of the review.

If I am reading these error bars correctly (would love some clarification if I'm not), very few of these empirical differences are even a standard deviation away from those generated without FedEx. These variance estimates make it *very* difficult (maybe impossible?) to evaluate the empirical value of this approach. Given that this paper is mostly framing / empirical in nature, and as far as I read the theoretical statement is there mostly to support the idea that this approach is reasonable, I would need to see more precise empirical results to raise my score.

This in fact raises a question: assuming my understanding of the error bars is correct, why weren't these numbers reduced simply by running *more* iterations--with different seeds, different client orders, etc (abstractly, taking more samples from the well-defined random variable, test error on dataset X)? Assuming that these samples are independent (and given the ability to fairly precisely specify the randomness in the algorithm, this does not seem like a crazy assumption to me), and the boundedness (and hence necessarily sub-Gaussian nature) of the metrics of interest, an out-of-the-box Chernoff-like bound should apply and the precision on the point estimates should be able to be driven as small as desired (given enough samples).

If my guess for the meaning of the error bars is incorrect (and they represent something like empirical std dev of independent runs of the algorithm), then I think the paper needs to be updated with some more detail of experimental setup, what precisely these numbers mean, how we should interpret the point estimates and the associated variances, probably a computation like the one just described, etc.

Either way, I think given the mostly-practical nature of this paper, there is some clarification to be done around the meaning of these numbers. Either these numbers should be driven down, or some explicit quantification of uncertainty around the point estimates should be provided. It is very difficult to evaluate the empirical contributions of this paper without this information, or with such wide uncertainty estimates.

**Time Spent Reviewing:**

6

---

> ### Author Response · Authors · 2021-08-10
> **Response to Reviewer onB2**
>
> Thank you for your thoughtful review; we answer your individual questions below. In particular, we hope to address what we believe is your main concern, which is the interpretation of the empirical results given large error bars for some settings. In summary: these numbers represent empirical standard deviations, not confidence intervals, and so will not necessarily tighten with more trials. The numbers as presented give a measure of the variation we might expect from running the different algorithms, and we believe the consistent improvement or at least matching by FedEx across all settings using SHA as a wrapper is indicative of its usefulness. Note that the improvement we observe is qualitatively similar to the improvement of weight-sharing methods in traditional NAS over SHA-like baselines (c.f. Tables 2 and 5 in [Li & Talwalkar (UAI 2019)](https://arxiv.org/abs/1902.07638)).
>
> We have also run further trials to include additional results computing 90% confidence intervals for the expected performance that we believe help confirm the submission-time results for the SHA wrapper. These details can be found at [this link](https://drive.google.com/file/d/1yzDLQIdA_ZNQ_6ngpuF-UA2Pd5GX3vRv/view), will be included in revision, and are discussed more below. We hope this information will make it easier to evaluate the empirical contributions of the paper.
>
> ### Responses to questions
> 1\. [*I find the formulation (1) a little strange--that is, there seems to be a suppressed dependence on the parameter w, or a suppressed integration over time. The output of Alg_a depends not just on T_j but also on w--so without express dependence here.*]
> - The parameter w is the output of Alg_a here. The function of a in the optimization problem in (1) is just running an optimization algorithm hyper-parameterized by a over the data and computing each task’s validation loss w.r.t. the parameter Alg_a returns.
>
> 2\. [*Is there a silent assumption in SGD_c that we are actually doing SGD on something? If not, maybe consider a rename.*]
> - We agree that it would improve clarity to use a different term like Loc_c since the on-device algorithm does not have to be SGD. This will be changed in our revision.
>
> 3\. [*There is an additional and somewhat subtle difficulty in evaluating hyperparameters in FL which could profitably be further highlighted in the challenges and baselines section. This is the time-dependence of evaluation metrics in many real world FL applications.*]
> - This is a good point and not a challenge our paper explicitly addresses, although the pluralistic approach for training models in the linked paper seems possible to extend to our tuning setting. We will discuss this in our revision.
>
> 4\. [*Nit: it seems like Alg1 and Fig 2 dont quite match--that is, it seems to me that Alg 1 corresponds to ‘last only’ and cant capture the other discount factors?*]
> - Yes, Algorithm 1 only captures last only. We can make the description more general by adding a line or two for state estimation.
>
> 5\. [*Another nit: a further natural baseline here (one that actually does seem relevant to me): how does performance scale here with respect to clients per round? This is relevant because of the calling-out of unnaturalness of only using last-round validation data--this becomes more reasonable as round size blows up. This would make the baselines much more compelling to me, to have a dataset of essentially unbounded scale (e.g. StackOverflow next-word-prediction) on which very large-scale training was analyzed.*]
> - Thank you for the suggestion. We will look into comparing with more clients per round.
>
> 6\. [*(this is potentially genuine confusion, as I am no expert in the NAS literature) It seems to me that the connection to NAS is a little unnatural. In my mind it seems easier to consider (6) and (7) direct from (2). So it seems like NAS and weight sharing...is only there to say ‘introducing this sampling/reparameterization procedure is not a crazy thing to do’? I think this is a reasonable framing, but I'm not sure it fits with the rest of the positioning of the paper--the rest of the paper positions a little too front-and-center if my understanding is correct.*]
> - Your understanding is correct: (6) and (7) do directly follow from (2) (as noted on line 190) and the connection to NAS indeed serves to contextualize the algorithmic approach. We believe the connection is valuable to include in order to better describe what may be an otherwise unnatural-seeming approach of sharing weights between configurations.
>
> 7\. [*Nit typo (L269): s/shws/shows*]
> - Thanks, fixed.
>
> 8\. [*The error bars for these table entries are really large. Do they represent an uncertainty estimate or an empirical standard deviation, and if the former why weren’t more experiments run? They make it difficult to evaluate the results.*]
> - The error bars represent the empirical standard deviation over three trials for each experimental setting, and so will not necessarily decrease with more trials. We will specify this more clearly in our revision. The variation of the random search results can be especially large because the lack of early stopping means few configurations can be tested, leading to possibly all of them being bad. Indeed the random search results in Table 1 for Shakespeare and CIFAR are likely too noisy to definitively determine significance but do strongly indicate that we should be using SHA instead. We do believe the general improvement using FedEx across most settings using SHA suggests the method should lead to practical improvement.
>
> - Nevertheless, we appreciate the usefulness of confidence intervals of the point estimates here. To do so we have run additional trials---for a total of 5 on Shakespeare, 10 on FEMNIST, 10 when using RS for CIFAR, and 6 when using SHA for CIFAR---in order to get somewhat smaller 90% confidence intervals. Note that running many more trials here is difficult because experiments are expensive---each Shakespeare trial requires 1 GPU-day---so it is not practical to obtain tight bounds using a distribution-free Chernoff-style approach. We opted for the more standard Student-t confidence interval construction and have uploaded the results at [this link](https://drive.google.com/file/d/1yzDLQIdA_ZNQ_6ngpuF-UA2Pd5GX3vRv/view). While even with this approach it is difficult to get two-sided significance (non-overlapping intervals), please note that in many settings---global SHA x i.i.d. Shakespeare, personalized RS x non-i.i.d. FEMNIST, global SHA x non-i.i.d. FEMNIST, global SHA x CIFAR, and personalized SHA x CIFAR---the entire 90% confidence interval around the mean FedEx performance is better than the empirical mean baseline performance. On the other hand, the reverse where the baseline is better does not occur in any setting. We hope this additional information will help you better evaluate the empirical contribution.

---

> > ### Comment · Reviewer_onB2 · 2021-08-19
> > **Increased score in response to clarification of empirical results**
> >
> > The authors have significantly clarified the meaning of the numerics in the paper in their response to my review--thus my score has correspondingly increased from a 4 to a 6.

---

### Decision · Program_Chairs · 2021-09-27

**Decision:**

Accept (Poster)

**Comment:**

This paper studies hyperparameter optimization in FL, which is often neglected but a crucial part of FL. The paper provides interesting ideas to tackle this challenging problem. After the discussion phase, the reviewers are all in favor of accepting the paper. I recommend acceptance. I suggest the authors revise the paper to address the reviewers' concerns.